# Chimeric Antigen Receptor Immunotherapy for Solid Tumors: Choosing the Right Ingredients for the Perfect Recipe

**DOI:** 10.3390/cancers14215351

**Published:** 2022-10-30

**Authors:** Luciano Castiello, Laura Santodonato, Mariarosaria Napolitano, Davide Carlei, Enrica Montefiore, Domenica Maria Monque, Giuseppina D’Agostino, Eleonora Aricò

**Affiliations:** 1Cell Factory FaBioCell, Core Facilities, Italian National Institute of Health, 00161 Rome, Italy; 2Research Coordination and Support Service, Italian National Institute of Health, 00161 Rome, Italy

**Keywords:** immunotherapy, chimeric antigen receptor, solid tumor, gene therapies

## Abstract

**Simple Summary:**

Despite the success in hematology, chimeric antigen receptor T cell therapies have shown, to date, unsatisfactory results in other clinical settings. A remarkable number of different CAR-based approaches have been developed, varying not only the specific antigen to be targeted, but also the type of cell to be modified, the costimulatory domain, and the additional signals incorporated to overcome solid-tumor-specific challenges. This variety of options has created a broad diversification of CAR approaches that, on one hand, may accelerate the identification of successful strategies, but on the other hand, may hamper the interpretation of clinical results and the overall advancement of the field. In this review, we present the most promising approaches under development and discuss their specific advantages and challenges to facilitate the identification of winning strategies.

**Abstract:**

Chimeric antigen receptor T cell therapies are revolutionizing the clinical practice of hematological tumors, whereas minimal progresses have been achieved in the solid tumor arena. Multiple reasons have been ascribed to this slower pace: The higher heterogeneity, the hurdles of defining reliable tumor antigens to target, and the broad repertoire of immune escape strategies developed by solid tumors are considered among the major ones. Currently, several CAR therapies are being investigated in preclinical and early clinical trials against solid tumors differing in the type of construct, the cells that are engineered, and the additional signals included with the CAR constructs to overcome solid tumor barriers. Additionally, novel approaches in development aim at overcoming some of the limitations that emerged with the approved therapies, such as large-scale manufacturing, duration of manufacturing, and logistical issues. In this review, we analyze the advantages and challenges of the different approaches under development, balancing the scientific evidences supporting specific choices with the manufacturing and regulatory issues that are essential for their further clinical development.

## 1. Introduction

The adoptive transfer of T cells genetically engineered to express chimeric antigen receptors (CAR) has conquered striking and long-lasting clinical responses in B cell malignancies and multiple myeloma, leading to the approval of six different CAR-based therapies in the last 5 years [1,2,3,4,5,6]. These results, together with a large body of preclinical literature, have fueled the development of an enormous amount of CAR-based therapies directed against solid tumors [7]. However, the rapid adaptation of current CAR technologies to solid tumors has to date failed the clinical challenge, highlighting the need to potentiate these adoptive therapies for the solid-tumor battlefield. A wide set of improvements have been hypothesized and are being evaluated in both preclinical and early clinical trials, but no consensus has been reached on the essential features needed to strengthen the clinical success of CAR-based therapies.

CAR-based therapies are the most complex advanced therapy medicinal products (ATMP) as they are the result of the introduction of a synthetic multi-domain protein (i.e., the CAR construct) into a population (more or less purified) of immune cells that need to be expanded and activated ex vivo before being administered to patients. In addition to the CAR construct, that provides the tumor antigen specificity, it is now clear that additional factors should be introduced into the immune cells to potentiate their anti-tumor activity in solid tumors. Moreover, the logistical and manufacturing constraints that emerged from the first CAR-based therapies in hematology have pointed to the need for novel manufacturing approaches [8]. Therefore, the technology behind CAR-based therapies lies in the intersection of different fields.

Herein, we aim to highlight the different approaches that are being developed in the CAR-based immunotherapies, including the manufacturing and regulatory issues that may surge during their clinical advancement. Single features of CAR-based therapies will be analyzed in view of their exploitation and increase the biological activity of CAR-based therapies against solid tumors.

## 2. The Challenges of Solid Tumors against CAR-Based Therapies (and Other Cell Immunotherapies)

Thanks to the knowledge collected in the last 20 years on cancer immunology, several immune escape mechanisms have been unraveled [9]. Most of these mechanisms are also behind the limited efficacy of CAR-based therapies in solid tumors [10,11] and should be bypassed to unleash CAR potentials.

One of the main challenges for CAR-based immunotherapy against solid tumors is the identification of a selective and reliable antigen to target. An ideal tumor antigen should be selectively expressed on tumor cells and not in normal tissues. In real life, tumor-restricted antigens are rare, and tumor-associated antigens (TAA) are often shared with nonmalignant tissues. Therefore, non-cancerous cells with low-level of TAA expression cannot escape the cytotoxic activity of the infused cells, the so-called on-target/off-tumor toxicity. Several cases of this phenomenon have been reported, the most notorious and fatal one occurred during the early development of anti-HER2 CAR-T, where the recognition of lung ERBB2+ epithelial cells by the infused cells caused massive cytokine release resulting in patient death [12] Of note, while in hematological malignancies several therapies allowing for the replenishment of immune cells can help in managing normal cells toxicity, it is difficult to contain healthy tissues damage in solid tumors.

The possibility of addressing this issue by fine-tuning the activation capacity of CAR cells based on the different levels of antigen expression between normal and cancer cells is hindered by the observation that solid tumors exhibit heterogeneous expression of tumor antigens per se. While CAR-based immunotherapy requires a certain, albeit variable, threshold of antigen expression to clear tumor cells, immune escape in solid tumors occurs through the downregulation or even loss of tumor antigens [13,14]. In fact, even oncogenic drivers can become unnecessary to tumor growth after tumor malignant transformation and be lost under selective pressure, thus impairing the long-term efficacy of CAR immunotherapies and increasing the risks of relapse.

Another issue to be overcome is that immune cell trafficking is impaired by several features typical of solid tumors. While in hematological malignancies the encounter between tumor cells and intravenously administered CAR-T is facilitated by being in the same peripheral blood compartment, solid tumors can be difficult to infiltrate. First, extravasation of immune cells is limited by the downregulation of specific adhesion molecules, such as ICAM-1 and VCAM-1, by tumor endothelial cells [15]. Second, malignant tissues are often dense due to enrichment in ECM components, such as hyaluronic acid and collagen that, in addition to creating a physical barrier against immune cells penetration, may inhibit T cells cytotoxic activity [16].

Finally, CAR-based therapies have to face and counteract the hostile solid tumor microenvironment (TME). Tumor cells express several molecules that exert inhibitory effects on anti-tumor immunity. Among them, immune checkpoint ligands play a pivotal role in regulating the cytotoxic activity and proliferation of infiltrating T and NK cells [17]. PD-1, LAG3, TIM3, TIGIT, KIR, CTLA, and VISTA are all targeted differently by cancer cells and other cells present in TME to silence anti-tumor responses [17]. Moreover, TME is populated by immune cells with immunosuppressive functions, such as regulatory T cells (Tregs), myeloid-derived suppressor cells (MDSCs), and tumor-associated macrophages (TAMs) [18]. Their pro-tumoral effect is mediated by the secretion of growth factors, chemokines, and cytokines that, in addition to supporting tumor cells proliferation, inhibit the cytotoxic activity of immune cells, including CAR cells of any type.

Additionally, solid tumor masses are characterized by a low level of oxygen (hypoxia) and restricted nutrient availability. These limitations strongly limit the vitality and anti-tumor activity of naturally occurring immune cells and, similarly, affect CAR-based therapies.

## 3. The Experience with CAR-Based Therapies in Solid Tumors

Initial attempts of using CAR-based T cells in solid tumors date even before those in the hematological fields [19,20,21,22,23]. These pioneer studies were based on the so-called “first-generation” CAR constructs, as they contained the sole CD3z domain in the cytoplasmic side. T cells modified with these CAR constructs were able to show antigen-specific cellular cytotoxicity, cytokine production, and favor T cell proliferation both in vitro and in animal models. However, the results collected in these clinical trials were largely disappointing [24], mostly due to limited proliferation and persistence of engineered cells.

Propelled by the results of second-generation CD19-CAR products against hematological malignancies, a second wave of clinical trials in solid tumors had started and continues to grow, as shown by the increasing number of active clinical trials worldwide [25]. Table 1 summarizes a selection of these studies, illustrating the variety of tumors and targets that have been explored. Despite the differences in clinical setting, tumor target, and type of construct, some common traits can be highlighted. In several studies, conditioning chemotherapy prior to CAR therapy improved the persistence of CAR engineered cells, in line with the observations in hematology and other adoptive cell transfer approaches [26,27]. The inclusion of a costimulatory domain in the CAR construct (i.e., second-generation CAR constructs) also resulted in longer persistence of engineered T cells. Moreover, an increase in serum cytokine levels, considered a CAR-associated clinical manifestation, was observed in several patients. Notably, in addition to some traits in common, some relevant differences with past experience in hematological CAR-based therapies emerged. For example, Haas at al. compared the expansion and persistence of anti-mesothelin CAR T cells with the results observed with anti-CD19 CAR T cells generated with the same construct (except for the recognized target) and manufacturing process. Notably, anti-mesothelin CAR T cells experienced a 10-fold less expansion and a significantly shorter persistence in vivo with respect to the CD19 counterpart. Interestingly, T cell expansion did not always correlate with increased serum cytokines after CAR cell infusion [28]. Even when T cells were engineered to be insensitive to the inhibitory signal transforming growth factor (TGF)-β, the observed CAR-associated increased serum cytokine levels did not consistently appear to be related to in vivo CAR T cell expansion over time [29]. This phenomenon was observed although engineered cell proliferation reached the same level seen in the hematological field. These evidences raise the question whether successful CAR-based therapies for solid tumors should closely mimic evidences collected in hematological CAR-based therapies or should follow a different path.

The analysis of TME performed in some of these clinical studies provided new information on the mechanisms developed by solid tumors to escape cytotoxicity of CAR cells. Heczey et al. reported that the blood of neuroblastoma patients treated with anti-GD2 CAR T cells showed a marked increase in M2-polarized macrophages after CAR cell infusion [36]. Whether this blood phenomenon reflects a parallel change in TME with increased M2 macrophages infiltration was not addressed, but can be considered highly reasonable. O’Rourke et al. analyzed glioblastoma samples resected 2 weeks after anti-EGFR CAR T cells infusion, and reported a significant increase in the levels of immunosuppressive molecules indoleamine 2,3-dioxygenase 1 (IDO1), TGF-b, IL-10, together with increased expression of PD-L1 and infiltration of Treg, thus suggesting that multifactorial immunosuppressive mechanisms are rapidly developed in situ following CAR T therapy [37]. On the other hand, the analysis of tumor samples after the infusion of CAR-engineered T cells also suggested the occurrence of epitope spreading [37,40,41]. 

In contrast to the anti-CD19 CAR experience, no clear observations can be highlighted regarding the costimulatory signal which is more functional against solid tumors. Of the eighteen studies reported in Table 1, eight employed the use of CAR construct containing CD28 costimulatory domain, eight had 4-1BB instead, and two studies used third- generation CAR construct containing CD28 costimulatory domain and 4-1BB or OX-40. Due to the differences in clinical setting, CAR target, and additional therapies in these studies, it is not possible to clearly establish the advantages and challenges of the different costimulatory signals for optimal anti-tumor activity of engineered cells. Similarly, no convincing evidences can be traced on which of the additional tools explored in these studies can boost CAR-based therapies in solid tumor settings. In two studies, inhibitors of the PD-1/PD-L1 axis were included without resulting in clear clinical benefit [28,41]. Similarly, in the only study where a dominant negative receptor for TGF-b was included into the CAR construct, high CAR T cell expansion and long persistence were recorded, without being associated with meaningful and durable clinical responses [29]. Finally, one study combined CAR engineering of T cells with CRISPR/cas9 mediated knock-out of TCRα subunit constant (TRAC) and PDCD1 genes that encodes for PD-1 [46]. These engineered cells infiltrated tumors, but still showed limited persistence.

More encouraging results were reported by Heczey et al. in patients with relapsed or refractory neuroblastoma using NKT cells [47]. In fact, the interim analysis of anti-GD2 CAR NKT cell trial of the lowest dose cohort showed high tumor infiltration of engineered cells and some indication of clinical benefit. Of the twelve treated patients, four had stable disease (SD), two had a partial response (PR), and one achieved complete response [48]. 

Therefore, despite these studies highlighted with some relevant preliminary observations, the optimal solutions to achieve effective CAR-based therapies against solid tumors still need to be identified. In the next paragraphs, the different parts composing CAR-based therapies will be discussed together with the different solutions under development.

## 4. The Cells to Engineer

One of the most debated aspects of CAR-based therapies is the choice of the cell population to be genetically engineered for the expression of CAR. Since CAR-engineered cells will represent the active substance of final product, the choice of cell population has a deep impact on large-scale manufacturing and “off-the-shelf” potential of the CAR therapy: Two critical points that will strongly dictate the future of CAR-based therapies for solid tumors in terms of clinical availability and economic sustainability (Figure 1) [49,50]. Moreover, since different immune cells exhibit different mechanisms of action and subsequent antitumor activity, each carrying advantages and challenges, the choice of the cell type will affect the in vivo clinical efficacy of CAR-based immunotherapy.

### 4.1. Bulk T Cells

The first preclinical and clinical studies, and currently the only CAR immunotherapies authorized for clinical use are based on bulk T cells. The only exception is lisocabtagene maraleucel, a CD19-directed CAR T cell product containing defined amounts of CD8+ and CD4+ cells.

CAR-based therapies using bulk T cells can rely on well-established safety profiles. With the exception of the on-target/off-tumor toxicity that is product specific, most of the safety concerns on bulk CAR-T relate to the occurrence of cytokine release syndrome (CRS) and immune effector cell-associated neurotoxicity syndrome (ICANS). CRS is triggered by the rapid and massive increase in serum cytokine levels upon T cell activation in vivo and may cause severe multi-organ dysfunction. Notably, the reported cases of CRS in solid tumors appeared to be milder and less frequent than the ones occurring in hematological malignancies [37,51,52]. Nevertheless, severe deadly cases of ICANS have been reported in patients with prostate cancer [53].

On the other hand, the antitumor activity of bulk T cells solely relies on CAR signaling. Therefore, the efficacy of bulk T cells engineered with current CAR constructs is hindered by the heterogeneity of antigen expression typical of some solid tumors, as well as by the antigen escape phenomena causing antigen loss.

As T cell phenotype and functionality are known to be significantly impacted by patient age, tumor burden, and previous cancer treatments [54,55], autologous CAR-engineered bulk T cells can significantly be impaired in their antitumor activity. One of the possible strategies to circumvent these limitations is the development of allogeneic CAR-T cells, in which the use of healthy third-party donors as a cell source guarantees better T cells fitness. Several genetically engineered allogeneic T cells are currently in the early phase of clinical testing on hematological malignancies and solid tumors. In addition to the clear advantages of these “off-the-shelf” products in terms of manufacturing standardization and timing, cost-effectiveness and potential efficacy, some significant challenges need to be carefully considered and evaluated before their widespread use. Among them, the increased risks of alloreactivity, in terms of graft-versus-host disease (GVHD), rejection due to immunogenicity, and poor persistence [56].

### 4.2. Specific T Cells Subsets

Despite the success of CAR-engineered bulk T cells in the hematological field, many groups have questioned whether other immune cells or specific T cell subsets may exert more potent antitumor activity and overcome some of their limitations [57,58,59,60]. In fact, CD3+ T cells comprehend a variety of subpopulations characterized by different cytotoxic activities, cytokine production, and proliferation potentials. Due to the differences in the phenotypic composition of the cell therapy tested in clinical trials, the potential impact of the different T cells subsects on bulk CAR-T efficacy and toxicity cannot be systematically assessed. Nevertheless, accumulating evidences point to the frequency within the infused product of a CD8+CD45RA+CCR7+ subset, closely resembling “T-memory stem cells” as a key factor for CAR-T in vivo expansion and persistence after the infusion [61]. Stem memory T cells are a rare subset of T cells firstly described by Gattinoni et al. in 2011 for being endowed with potent stem cell-like ability to self-renew and the multipotent ability to originate central memory, effector memory, and effector T cells [62]. Tscm cells are currently stimulating a big interest for their possible exploitation in adoptive immunotherapy, while facing the difficulties related to the ex vivo manipulation and expansion of this rare cell subset. Recently, preclinical data reported the superiority of Tscm CAR-T cells (CD4+CD8+CD62L+CD45RA+) with respect to bulk T cells (CD4+CD8+) in terms of antitumor activity and expansion capacity in xenograft mouse models of leukemia [63] and lymphoma [64]. Furthermore, Tscm cells were less prone to induce CRS in mice, thus holding promise for a more effective and safer cell product [63].

Other T cell subsets undergoing preclinical and clinical evaluation as reliable sources for CAR-based therapies are two rare populations at the intersection between adaptive and innate immunity: Gammadelta (γδ) T cells and invariant natural killer T (iNKT) cells. γδ T cells bear a TCR made of a γ chain and a δ chain and are more abundant in epithelial sites, such as skin, tongue, and intestine, than in peripheral blood [60]. Since γδ T cells recognize their non-peptide phosphorylated metabolic intermediates called “phosphoantigens” at the earliest signs of tumor cell modification and independently from MHC restriction, they can be considered part of the innate immune response. Nevertheless, γδT can create an immunological memory, such as the adaptive immune response. Another advantage of γδT cells is that these cells secrete lower levels of proinflammatory cytokines with respect to αβT cells, thus are supposed to be less prone to induce CRS. Moreover, they are endowed with an increased tropism for tumor microenvironment. Altogether, these features combined with the reduced risk of GVHD, make them particularly suited for allogeneic immunotherapy and “off-the shelf” CAR-T generation. One of the major bottlenecks of their clinical application is the large-scale expansion of this scarce cell population. CAR-modified expanded γδ T cells were tested against CD19- or GD2-expressing tumors and other tumor antigens in vitro and in clinical trials [65]. In hematological setting, CAR-γδ T cells showed the ability to cross-present tumor antigens to αβ T cells in vitro, as well as antigen-dependent and independent cytotoxicity. More recently, the feasibility and antitumor activity of allogeneic Vδ1, the subset more frequent within TIL, was reported in a mouse model of hepatocellular carcinoma [66], and a similar approach is now undergoing clinical trial testing [65].

The iNKT cells are a subset of lipid and glycolipid-reactive T lymphocytes that express an invariant TCRα chain rearrangement paired with a restricted repertoire of TCRβ chains [67]. Upon recognition of glycolipid antigens expressed on CD1d molecules, iNKT cells rapidly secrete immunomodulatory cytokines and directly mediate cell cytotoxicity through the secretion of perforins and granzymes. In addition to this direct natural anti-tumor activity, iNKT cells can reshape TME by polarizing TAM and MDSC toward immune-stimulating cells. The iNKT cells actively localize to tumors in response to CCL2 and CCL20 [68]. Therefore, they represent a valuable cell source to overcome some solid tumor pitfalls. Moreover, iNKT cells have been successfully isolated from peripheral blood and engineered to express CAR both in preclinical and clinical studies [47,57,69,70,71]. As mentioned in the previous paragraph, anti-GD2/IL15-expressing iNKT cells have shown promising results in neuroblastoma patients with high tumor infiltration and evidences on clinical responses [47,48].

### 4.3. Natural Killer Cells

NK cells play a crucial role in the innate immune response, since they are endowed with the ability to differentially sense healthy cells vs. cells showing signs of stress as a consequence of infection, damage, and malignant transformation. Circulating NK cells represent 5 to 15% of human leukocytes and are armed with an array of receptors, whose delicate balance determine NK ability to recognize and rapidly act against malignant cells without prior sensitization. Activated NK cells can exert their cytotoxic activity through perforin and granzymes granules release or use their FcγRIIIA (CD16) receptor to recognize antibody-coated cells and activate antibody-dependent cellular cytotoxicity (ADCC) and cytokine production.

Some features of NK cells make them particularly promising for CAR-based therapy. When compared to bulk T cells, CAR-NK showed in both preclinical and clinical settings a better safety profile, with minimal cytokine release syndrome or severe neurotoxicity [72]. Moreover, the ability to kill tumor cells by CAR-independent mechanisms represents an extra weapon against tumors with an uneven expression of CAR-targeted antigen.

From a safety point-of-view, the short half-life of NK cells in vivo is a “mixed blessing” for CAR-NK therapy: While it guarantees limited long-term toxicity, it also demands repeated administrations to achieve clinical durable response. Studies are currently exploring the possibility to prolong CAR-NK persistence in vivo by armoring them with cytokines-encoding genes [72] or by adding to the manufacturing process a pre-activating step with IL-12/15/18 to induce differentiation into cytokine-induced memory-like NK cells [73].

Since NK cells have a very limited risk of graft versus host disease when used in allogeneic settings (as confirmed in [72]), CAR-NK can be manufactured as “off-the shelf” products from several allogeneic NK sources, such as donor PBMC, immortalized cell lines, and cord blood induced pluripotent stem cells (iPSCs). Some groups have explored the well-established NK cell lines NK92 as a cell source in early clinical testing, and collected promising results in patients with leukemia [74]. However, the use of NK92 requires irradiation of CAR-NK before infusion as a safety measure to limit the oncogenic potential of this transformed cell line, thus further shortening the in vivo persistence of the infused cells. The use of donor PBMC combines the advantages of increased safety and maximum cytotoxic activity, but holds some difficulties in expansion and transduction efficacy. Efforts are ongoing to identify best culture conditions for high expansion and activation [75,76].

### 4.4. Macrophages

Macrophages are innate cells of the immune system that depending on the external signals polarize into specific subsets and exert pro-inflammatory (M1 macrophages) or anti-inflammatory/tissue repairing effects (M2 macrophages). They represent the most abundant immune cells subtype within the tumor. M1 macrophages are crucial for immune response against infected and malignant cells, such as phagocyitic and cytotoxic activity, antigen presentation, as well as cytokines and chemokines secretion for immune cells recruitment. However, macrophages resident at the tumor site, so-called tumor associated macrophages, are often polarized toward an immunosuppressive phenotype closely related to an M2 phenotype. The possibility to exploit and potentiate the innate phagocytic activity of macrophages in immunotherapy strategies was first exploited with the development of a CD-19-directed CAR construct bearing phagocytic receptor signaling domains (CAR-P) [77]. Since then, another strategy involved the creation of CAR macrophages secreting metalloproteinases to target the ECM, showing antitumor efficacy in a mouse model of HER2+ breast cancer [78]. Klichinsky et al. established a CAR-M platform capable of polarizing engineered macrophages toward an M1 phenotype, with improved capacity to direct anti-tumor activity as well as perform cross-presentation and T cell costimulation [79]. A clinical study is currently testing the safety of autologous HER2-directed CAR-M obtained with this technology against HER2 expressing solid tumors (NCT04660929).

Due to their ability to sense and reject foreign antigens and genetic material and their limited proliferation potential, manufacturing of CAR-M still poses several challenges. 

## 5. The Gene Engineering Tools

A broad array of technological solutions for cell gene engineering are now available [80]. Viral vectors have longer and wider clinical track record, whereas transposon-based editing and crispr/cas9 approach have emerged in the last decade. Each tool has its own intrinsic advantages and challenges.

### 5.1. Cell Transduction Using Viral Vectors

Viral vector-based gene engineering represents the most consolidated approach for gene therapy. Viral vectors used for CAR engineering belong to three major families: Gammaretroviral, lentiviral, and adenoviral vectors. While the first two allow for the integration of the construct into the cell genome for stable expression in cell progeny of proliferating cells and represent the choice solution for T and NK cells, adenovirus vectors are mainly indicated for expression in low-proliferating cells and have been only used to engineer macrophages [81].

Four of the six FDA/EMA approved CAR-based therapies (i.e., Abecma, Breyanzi, Carvikty, and Kymriah) utilize lentiviral vectors for CAR integration, whereas the remaining two, Tecartus and Yescarta, rely on gammaretroviral vector for CAR delivery. The major differences of the two families of vectors are in regard to when and where they integrate into the host genome [80]. DNA integration of gammaretroviral vectors take place only during cell division, whereas lentiviral vector can integrate even in non-proliferating cells. Moreover, gammaretroviral vectors show an integration bias for promoter regions and growth control genes [82], while DNA integration of lentiviral vectors usually occurs in the introns of transcriptionally active genes. As a consequence, gammaretroviral vectors have shown a higher genotoxicity in hematopoietic stem cell gene therapy. However, no oncogenic T cell transformation has been observed to date with gammaretroviral transduced T cells [83]. Nevertheless, lentiviral vectors have collected a wider preference for CAR transduction by the research community for safety concerns [84].

Both families of vectors show high transduction efficiency in T cells, in NK cells, and to a limited extent in macrophages. Their large packaging size enables the delivery of CAR construct in addition to multicistronic transgenes, thus representing valuable tools to incorporate additional modifications to the cytotoxic cells [84]. In fact, vectors containing 8–9 kb can be generated with both backbones at good titers, whereas longer constructs have logarithmically reduced titers [85].

Despite the large packaging size and the high transduction efficiency, viral vectors show some challenges as compared to other gene therapy platforms. First, GMP-compliant manufacturing of these vectors is complex and expensive. Second, safety concerns dictate deeper characterization both of the vector used and of the transduced cells, thus further increasing the cost of manufacturing.

### 5.2. Transposons for Gene Therapy of CAR

In nature, transposons are genetic elements that can change their positions within a genome thanks to the activity of a transposase protein. This enzyme, encoded within the transposon sequence, recognizes specific genome sequences, so-called the terminal inverted regions. In addition, it mediates the excision of the element from the genome and allows its integration into another locus containing the same terminal inverted regions [86]. Since they combine the desired stable integration and transgene expression (typical of integrating viral vectors) with lower immunogenicity and reduced costs for GMP manufacturing [80], transposon-based gene therapy systems represent a valuable alternative to viral vectors. Notably, while the piggyBac transposon system has an integration profile similar to the one of gammaretroviral vectors (with the same risk of genotoxicity and related safety concerns), the sleeping beauty (SB) transposon system has a significantly safer profile of viral vectors as it mediates construct integration in a close-to-random fashion [84].

In contrast to viral vectors that are inherently capable of entering into target cells, transposon-based vectors require cell electroporation to allow for the DNA to enter into target cell nucleus. While short constructs can be easily electroporated into target cells, larger constructs enter the cell with lower efficiency and can reduce cell viability due to cytotoxicity phenomena [87]. Therefore, some additional optimization steps are required to maximize viability and efficiency of SB-based multicistronic constructs for CAR engineering.

A variety of approaches utilizing the SB transposon system have been described. The classical and simplest two vector-based approaches (where one vector contains the CAR construct with the terminal inverted regions and one vector encodes the transposase) has been already used in CAR clinical trials in the hematology and showed high CAR integration efficiency, a good safety profile, and clinical response [88,89]. More recently, a novel approach that utilizes one mRNA encoding for an high-active transposase and one minicircle DNA as the vector to be integrated has been developed [90]. This strategy offers the advantage of ensuring a fast and transient expression of transposase, thus limiting the genotoxic risk of transgene remobilization. Moreover, the reduced length of minicircle DNA increases the efficiency of electroporation and results in higher post-electroporation cell viability. This solution has been already implemented in CAR-based gene therapy products and is currently being investigated in a clinical trial in multiple myeloma patients [91].

In conclusion, although the safety profile of viral vectors relies on a significantly larger number of patients as compared to newly developed engineering systems, the continuous advancement and refinement of more efficient transposases and the encouraging clinical results collected to date make the SB transposon system a valuable and practical solution particularly promising for CAR cell engineering.

### 5.3. Gene Editing Using CRISPR/cas for CAR Engineering

CRISPR/cas technology is revolutionizing the gene engineering field permitting an unprecedented manipulation of genomic DNA [92]. Discovered only 10 years ago, this technology takes advantage of an ancient bacterial/archea defense mechanism against viruses and plasmids [93]. Basically, CRISPR/cas can recognize specific genomic sequences and catalyze their cleavage. Depending on the specific type of cas protein, a single-stranded nick, staggered or blunt double strand breaks (DSB) can be generated [92]. Currently, the CRISPR/cas9 protein is the most studied for gene engineering of cells for clinical use. In fact, CRISPR/cas9 generates DSB in specific genes of interest (as indicated by the guide RNA, gRNA). Then, these DSB can be used for gene deletion or gene insertion, depending on the DNA repair mechanism involved. In fact, the majority of DSB are resolved by the error-prone non-homologous end joining (NHEJ) pathway. This pathway introduces small insertions or deletions that result in frameshift mutations and premature stop codons of the gene of interest. The other repair mechanism is the homology-directed repair (HDR) that enables transgene integration at the break site when the transgene sequence (as linear dsDNA) is provided together with flanked homology arms [92]. Notably, despite the fact that improved in silico tools have been developed to design/select optimal gRNA, off-target modifications can occur, warrying the deep characterization of the safety profile of selected gRNA.

As mentioned, CRISPR/cas technology has already been used in one clinical trial in solid tumors expressing mesothelin [46] targeting PD-1 and TRAC genes to block tumor inhibitory signaling (PD-1) and to reduce the risk of autoimmune responses as a consequence of PD-1 KO (TRAC), but no improvements in terms of persistence or clinical benefit was registered. Other than being limited to inhibitory pathways, the targeted gene deletion has also been explored to generate “universal” CAR T cells by deleting the TCR receptor. This approach has already been tested in hematological clinical trials by means of the older gene editing tool, TALEN [94].

The use of CRISPR/cas to insert the CAR construct in specific regions of the genome has been explored in pre-clinical studies aiming at introducing the CAR in replacement of natural TCR, thus aiming at a more physiological CAR expression [95]. Notably, the authors observed that this genetic modification induced a decrease in T cells antigen-independent activation status, the so-called tonic signaling. Of note, lower tonic signaling resulted in reduced cell differentiation and exhaustion, thus increasing the potency of the engineered cells. Alternatively, it has been proposed to insert the CAR construct in genes that could have an inhibitory signal, such as the abovementioned PD-1 [80].

Although CRISPR/cas genetic manipulation has already reached the clinical arena with the cas9 protein derived from *Streptococcus pyogenes*, other more advanced cas protein-based approaches, natural or synthetic, have shown superior specificity and reduced off-target modifications [92,96]. These novel tools will probably replace former ones for targeted gene engineering of CAR-based therapies; notwithstanding, only larger clinical studies will reveal their long-term safety profile. 

## 6. Designing the Best CAR Construct against Solid Tumors

As schematically shown in Figure 2a, a typical CAR contains: The antigen-recognition domain (usually a single-chain variable fragment, scFv), the hinge domain (that connects the scFv to the transmembrane portion), one transmembrane domain, and the intracellular signaling region that is usually composed of one or more costimulatory domain(s) and a CD3ζ T cell activation domain. While most of the research has focused on the scFv and on the costimulatory signals, changes in the hinge and transmembrane domains also affect cell activity, and thus should be more carefully characterized [97]. Moreover, it has been clearly shown how the CAR expression level affects cell proliferation and antitumor activity through the antigen-independent tonic signaling, thus underlining the relevance of CAR promoter in regulating the activity and the toxicity of CAR-based therapies [98,99]. Therefore, it is reasonable to hypothesize that each CAR domain should be differently chosen depending on several variables, including the tumor antigen expression level, the affinity and the avidity of the scFv and, of course, depending on the cell to be engineered.

### 6.1. Tuning the Expression Level of CAR

In contrast to gene therapies that aim at replacing a disease-causing gene where broader ranges of transgene expression levels can still result in clinical benefit, the introduction of CAR requires a finer tuning as its expression dictates the fate of engineered cells, their long-term efficacy, and their toxicity. While most of the developed CAR rely on strong promoters to ensure a constitutive high transgene expression, several clues support modifications to this paradigm. 

In T cells, it has been shown that the high expression of CAR can result in tonic signaling, a constitutive signaling that can exert different effects depending on the costimulatory signal contained in the CAR. In fact, CD28-driven tonic signaling drives T cell differentiation and exhaustion, thus strongly limiting anti-tumoral activity [98]. Although tonic signaling of 4-1BB-containing CAR was shown to be associated with reduced exhaustion as compared to CD28-containing CAR [98,100], 4-1BB-driven tonic signaling can induce T cell apoptosis [101]. Moreover, lowering CAR expression using weaker promoters can result in increased in vivo anti-tumor activity [101,102]. Less is known regarding how CAR expression can affect the activity of engineered NK cells or macrophages. How CAR signaling interacts with the complex array of stimulatory and inhibitory signals that regulate NK cells activity and to what extent the CAR expression level affects NK cell anti-tumor activity have been poorly analyzed, but are of extreme importance. The expression level of CAR on macrophages is expected to play a less relevant role considering the reduced proliferation and persistence of macrophages [79], but specific studies characterizing the effect of CAR signaling on the development of “memory”-like traits in macrophages should be encouraged.

A fine tune of CAR expression should also be faced with one of the major hallmarks of solid tumors: Hypoxia. Considering the activation of specific transcription programs during hypoxia, several groups exploited the advantages of including hypoxia-driven promoters into the CAR construct [103,104,105]. This approach may have wide applications in limiting the on-target/off-tumor toxicity, thus opening a wider choice of tumor antigens to be targeted by CAR. This strategy has not yet reached clinical testing, but it could represent an added tool for CAR-based therapies in solid tumors.

### 6.2. Defining Which Antigen to Target and How

As mentioned in Section 2, the selection of specific antigens against which redirect immune system is a major challenge for CAR-therapies against solid tumors, that often lack a universal, specific, and unique marker and share the risk of undergoing antigen loss. In particular, the expression of TAA by non-cancerous cells do represent a major safety issue for CAR-based therapies, as strong on-target/off-tumor toxicities can be generated [12,106]. To bypass this toxicity, it has been shown that selecting low-affinity scFv strongly reduces or completely eliminates the on-target/off-tumor reactivity of engineered cells while sparing their anti-tumoral activity [107,108,109]. Interestingly, it has been hypothesized that the reduced affinity of CAR may also positively impact T cell activation and differentiation by mimicking more closely the dynamics of TCR-MHC interactions [109].

In parallel, CAR-based therapies for solid tumors can strongly benefit from targeting multiple TAA. On one hand, this approach has been explored to overcome the heterogeneous expression of one single TAA. Using tandem CAR or utilizing constructs coding for multiple fully-functional CAR, several groups have shown a broader ability of engineered cells to recognize and kill tumor cells [110,111]. For example, Hedge et al. reported preclinical data on engineering T cells to express CAR against HER2 or IL-13Ra2: Upon encountering both antigens, a superadditive T cell activation was observed together with reduced antigen escape mechanisms in preclinical models. This approach achieved high response rates even in patients with high tumor burden and aggressive disease in the hematology fields and is currently under evaluation in solid tumor clinical trials [25,112,113]. On the other hand, the targeting of multiple antigens can be utilized for redirecting the antitumor activity of engineered cells only to cells expressing both antigens. This has been achieved by engineering cells with conditional-CAR or utilizing one CAR lacking the costimulatory domain with a chimeric costimulatory domain (see Figure 2b) [114,115,116,117]. Notably, the above-described approaches apply more to engineering T cells than other cell types that have an innate ability to recognize tumor cells (e.g., NK cells and γδ T cells) or can reshape TME (such as, macrophages and iNKT cells).

### 6.3. Choosing the Costimulation Signal

In nature, immune cell activation is usually the result of several signals that cooperate to trigger finely-tuned downstream signaling cascades, ultimately leading to the induction of cytotoxicity against the target cells, metabolic rewiring, and the induction of differentiation/proliferation processes [118,119,120]. Therefore, it is not surprising that an ongoing debate on which the signal is sufficient for optimized anti-tumor activity of engineered cells has continued since the first studies of second-generation CAR [24,121,122,123,124]. Due to historical reasons, the vast majority of preclinical and clinical evidences have been collected on costimulation signals for CAR T cells and, in particular, on CD28 and 4-1BB- derived costimulatory domains. All the approved CAR-based therapies do contain CD28 or 4-1BB costimulatory domain in their constructs.

CD28 is a costimulatory receptor that coclusterizes with TCR in the immunological synapses and potentiate TCR signaling, enabling large cytokine production, cell proliferation and differentiation, and metabolic rewiring. 4-1BB is one of the costimulatory receptors that is rapidly expressed by activated T cells [24]. Moreover, its signaling enhances T cell proliferation, cytokine secretion, and cytolytic activity, but it has a more profound effect in favoring the development and maintenance of memory CD8 T cells [125]. However, although the detailed signaling and functional role of both native proteins has been well characterized, it is not possible to extrapolate these effects onto CAR containing those domains as timing, kinetic, and spatial aspects strongly differ [126]. Despite the large body of preclinical and clinical data, no clear conclusions can be inferred regarding which costimulatory domain is preferable. As clearly reviewed by Cappell and Kochenderfer [24], most preclinical studies point to CD28-containing CAR T cells as producing higher levels of cytokines, whereas 4-1BB-containing ones have greater persistence. Similarly, clinical evidences highlighted higher percentage of patients experiencing cytokine release syndrome after receiving CD28-containing CAR T cells and longer persistence of engineered T cells was observed in clinical trials with CAR T cells containing the 4-1BB costimulatory domain, while no clear differences in efficacy can be established [24]. Some preclinical results point to CD28 costimulation as more useful in the solid TME, as the large cytokine production can counteract the immunosuppressive milieu of TME [127]. Analyzing the inhibitory effect of Treg cells to CAR T cells harboring CD28 or 4-1BB costimulation domain, Kegler et al. showed a reduced inhibition of CD28-containing CAR cells in vitro and in vivo, possibly due to the high cytokine production.

Additionally, it has to be noted that confusing data exist on the combination of both CD28 and 4-1BB costimulatory domains into the same construct (the so-called third-generation CAR) [128]. In fact, while CAR T cells engineered with both costimulatory domains have strong anti-tumor activity in vitro, clinical evidences have not shown striking advantages over second-generation CAR therapies. Of note, one recent report suggests a superior activity of third-generation constructs containing 4-1BB signaling domain together with a mutated CD28 one [24,129]. Therefore, it is not possible to rule out that more advanced third-generation CAR constructs will be among the key elements for the success of CAR-T cells in solid tumors.

Significant limited knowledge has been collected on the optimal costimulation signals needed for CAR-NK and CAR-M. Although good clinical results were observed in lymphoid tumor patients treated with NK cells engineered with a CD28-containing construct [72], several preclinical studies suggest that NK cells engineered with “natural” costimulatory signals outperform those engineered with T cell-derived ones [130,131,132]. In these studies, in fact, by replacing CD28 or 4-1BB domains with 2B4, DNAM1, DAP10 or DAP12, engineered NK cells showed improved in vitro and in vivo anti-tumor activity in animal models.

### 6.4. The Additional Signals to Overcome Solid Tumor Resistance

As already mentioned, several groups are exploring the added value of blocking by different means the PD-1/PD-L1 axis to unleash T cell activity [133,134,135,136]. Disrupting PD-1 gene, furnishing a dominant negative PD-1 receptor, secreting PD-1 blocking scFv or furnishing a chimeric switch receptor (i.e., a receptor that upon ligand interaction transmits an intracellular signaling of an immune activating receptor) have all increased the anti-tumor activity of engineered T cells in preclinical setting. However, more controversial is the real advantage of this approach over combining CAR cells administration with pharmacological treatment with the PD-1/PD-L1 inhibitors for the worrisome considerations regarding the possible increased toxicity of these engineered cells. Similar approaches have been proposed also against other immunosuppressive signals [29,137].

Other groups are optimizing the ability of engineered cells to infiltrate the physical barrier of ECM that strongly limit infiltration of immune cells into solid tumors. It has been reported that the ex vivo expansion of T cells significantly reduces their capacity to degrade ECM. To bypass this point, Caruana et al. showed that an improved ability to degrade tumor ECM can be restored in T cells by inducing the expression of heparinase [138]. Other strategies exploit tumor ability to attract suppressive immune cells through the chemokine-receptor axis to increase CAR cells tumor infiltration. In recent years, the attempts performed with chemokine receptors CCR2, CCR4, CXCR2, CXCR3, CXCR4, and CX3CR1 (reviewed in [139]) showed promising results in preclinical settings. However, CXCR2 and CCR4 are the only chemokine receptors undergoing clinical trials to date. To easily bypass this bottleneck, several groups are exploring the locoregional delivery of CAR cells (reviewed in [140]). Data collected in clinical trials (mainly phase I) indicate that this can be considered a feasible strategy to improve CAR cell penetration into tumors while reducing the risks of systemic toxicity. The results of many clinical trials currently ongoing will further assess the safety and efficacy of this strategy.

Another line of promising direction is the inclusion of cytokine genes into the construct that can act by autocrine signaling to potentiate CAR-cell activation and by paracrine signaling activating other immune cells within TME [141]. Several studies have analyzed the addition of one or two cytokines into the construct. The most explored cytokines are IL-7, IL-12, IL-15, IL-18, and IL-23 with different rationales. CAR-T cells secreting IL-12 showed increased antitumor activity against ovarian cancer xenografts with prolonged cell persistence, higher levels of IFNγ, resistance to Treg immunosuppression, and expansion of natural antitumor responses through the activation of infiltrating macrophages [142,143]. Similar observations were also recorded with IL-18 expressing CAR T cells [141]. IL-7 or IL-15 are explored for their ability to increase the persistence of CAR engineered cells (T cells or NK cells) and preserve a less differentiated Tscm phenotype. Of particular interest are the combination of IL-7 with the C-C chemokine ligand 19 (CCL19) and CCL21 [144,145]. Both approaches resulted in an increased antitumor activity thanks to the higher tumor infiltration not only of the engineered cells, but also of dendritic cells.

Finally, to counteract the harsh TME, some groups have analyzed the added value of engineering proteins for metabolic resistance of engineered cells. The low amount of key nutrients, such as arginine, strongly impact the ability of T cells to mount a strong response. For this reason, Fultang et al. engineered CAR T cells with the arginine resynthesis pathway enzymes ASS and OTC [146]. CAR T cells with this metabolic engineering showed preserved proliferation and anti-tumor activity also in low-arginine settings. On the other hand, Ligtenberg et al. showed the direct and indirect benefits of adding catalase enzyme to counteract the increased levels of reactive oxygen species (ROS) often observed in TME [147]. In fact, the expression of catalase resulted not only in the preservation of the activity of engineered cells, but also of infiltrating T and NK cells.

## 7. Manufacturing and Regulatory Aspects to Consider

Prior to reaching marketing authorization or even phase I clinical trial (depending on the local regulatory agency), CAR-based therapies have to be manufactured according to the good manufacturing practice (GMP). GMP requirements include the validation of manufacturing processes, including ancillary and starting materials, and the establishment of a quality control strategy across the whole process from the collection of starting cells up to the infusion of final product. Both aspects can represent challenges for complex CAR-based therapies. As the scientific research of more promising CAR-based therapies against solid tumors rapidly progresses, big efforts on large-scale manufacturing and process optimization are needed to ensure the rapid transfer of the collected knowledge into the clinical routine.

Other issues hamper the clinical translation of potential candidates during all phases of CAR product development. During the collection of non-clinical data to be included in the ATMP dossier, there is limited availability of suitable preclinical and animal models to predict the safety and efficacy of CAR-based therapies. In fact, in addition to the inter-species differences affecting reliable toxicology evaluations, no animal model can faithfully recapitulate the complexity of tumor-immune cells interactions occurring in TME of solid tumors. More advanced humanized mice and 3D organoid in vitro models are under development, and should be carefully evaluated and selected.

During CAR product clinical development, the setup and maintenance of GMP-compliant manufacturing processes and relative QC strategies may be laborious and expensive, and a variety of logistical issues have to be managed. In this light, one of the major lessons learned after the FDA/EMA marketing authorization of the first CAR-T cell therapy is that the manufacturing of autologous products in centralized facilities has clear limits of scalability [148,149]. This issue is pushing towards two directions: The design and development of allogeneic/universal products, and the implementation of CAR-based therapies with minimized manufacturing. These two directions strongly differ in terms of the type of product that can be manufactured, the regulatory requirements it has to fulfill, and the manufacturing capacity. Allogeneic/universal CAR-based therapies hold the promise to strongly decrease the burden of manufacturing, the costs of production and quality control, and could be used as “off-the-shelf” therapies. These types of products can start from healthy donor samples, have an higher complexity in terms of gene manipulations (with the consequent complications of QC for safety assessment), should rely on large cell expansion processes allowing for the generation of high number of cells that preserve the functionality, and can be manufactured in centralized facilities, similarly to traditional drug manufacturing. This approach can be easily envisioned with cells more suited for allogeneic use (such as NK and specific T cell subsets), but with appropriate gene manipulations, this approach has been applied to bulk T cells, as well [76,148,150]. 

On the other hand, there is a growing interest in point-of-care (POC) manufacturing, although the term is often mistakenly used in academic clinical trials with one or few manufacturing sites [151]. POC manufacturing has the clear advantages of eliminating the need for transporting patient cells to centralized facilities for manufacturing and back to the administration site, but poses many challenges in terms of standardization and reproducibility. In fact, it should represent a novel model for decentralized manufacturing of CAR therapies and other products. Despite the fact that a specific regulatory framework for this approach is still lacking, an initial consultation has been opened by the UK NHRA and both FDA and EMA are planning to carry this out in the near future [152,153]. Therefore, POC manufacturing should rely on automated cell processing devices and isolators, highly-standardized manufacturing procedures and quality control assays, as well as robust pharmaceutical quality systems at each POC. As a result, simplified methods of manufacturing in terms of gene manipulations and in product turnaround time will maximally benefit POC manufacturing.

## 8. Conclusions

Since solid tumors are equipped with an ample armamentarium to evade and resist anti-tumor immune responses, a large variety of approaches to improve CAR-based therapies against solid tumors are under development and evaluation. Although the results collected to date were disappointing, clinical translation of the most recent advancements is expected to show some improvements. An accurate evaluation should guide the choice of cell types which are more suitable for exerting their antitumor activity in a specific clinical setting (i.e., NK to face possible antigen loss, Tscm for longer persistence) or more appropriately, for the preparation of allogeneic CARs (γδT, iNKT, NK). The use of refined CAR constructs, including the proper costimulatory signal, will help in balancing the optimal CAR expression and signaling with additional features specifically designed to overcome the harshness of TME. Adding the expression of chemokine receptors to improve tumor infiltration, of proteins interfering with the immune checkpoint axis, of immunomodulating cytokines, and/or of proteins ensuring a better metabolic fitness are a few examples of the possibilities to empower CAR-based therapies against solid tumors. Given the large amount of factors taking place on CAR product design and on patient tumor heterogeneity, it is reasonable to expect that employing multiple combinations of these features can broaden the chances to collect higher and longer response rates in treated patients. However, only joint coordinated efforts tackling the many variables involved with a multidisciplinary approach will definitively expedite the identification of the best solutions to be included in the CAR-based therapies.

Recent technical and technological advancements, such as the automation of manufacturing and the development of precise and safe gene engineering tools, can potentially pull forward novel concepts applicable to CAR-based therapies. Some of the most recent discoveries and developments have been discussed in this review, but surely additional others still have to be developed. Additional efforts should be placed on the development of fast in vitro assays for the evaluation of safety of engineered cells. Finally, pharmaceutical companies and regulatory agencies have to continue developing the layout for innovative and sustainable manufacturing of this complex type of drug to ensure that once the efficacy of novel CAR-based therapies will be proved, its rapid global clinical adoption can be foreseen.

## Figures and Tables

**Figure 1 cancers-14-05351-f001:**
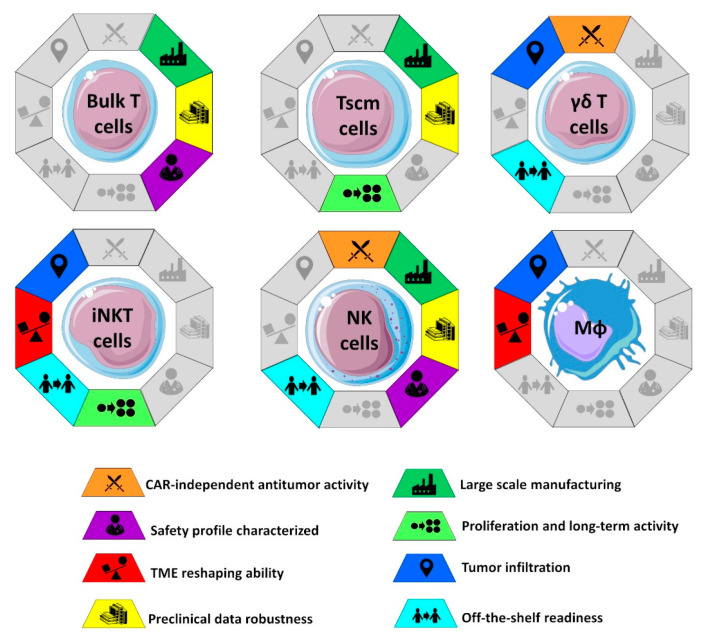
Characteristics of the different cells engineered with CAR constructs in solid tumors. As discussed in the main text, some of these characteristics can be acquired through additional gene manipulation. The Figure was partly generated using Servier Medical Art, provided by Servier, licensed under a Creative Commons Attribution 3.0 unported license.

**Figure 2 cancers-14-05351-f002:**
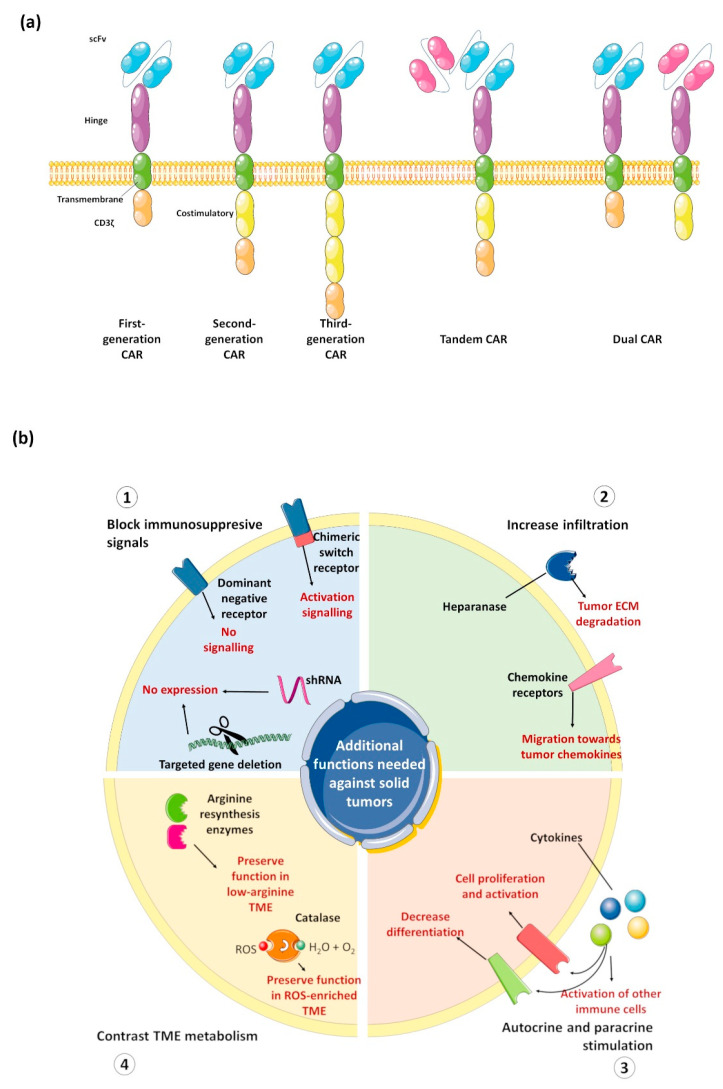
CAR designs against solid tumors. (**a**) Typical design of most common CAR constructs, from historical first-generation CAR construct to constructs simultaneously targeting two targets; (**b**) schematic representation of major strategies explored to overcome resistance of solid tumor to CAR-based therapies. The Figure was partly generated using Servier Medical Art, provided by Servier, licensed under a Creative Commons Attribution 3.0 unported license.

**Table 1 cancers-14-05351-t001:** Selected clinical trials utilizing second- and third-generation CAR T cells in solid tumors.

Disease	Trial Phase	No. of pt	Target	CAR Construct	Additional Strategies	Ref.
HER2+ solid tumors	I/II	19	HER2	CD28, CD3z	None	[30]
Liver metastases	I	6	CEA	CD28, CD3z	None	[31]
Non-Small Cell Lung Cancer	I	11	EGFR	4-1BB, CD3z	None, Cy * alone or Cy with additional cytotoxic drugs	[32]
Biliary tract cancer	I	19	EGFR	4-1BB, CD3z	Cy/nab-paclitaxel	[33]
Metastatic colorectal cancer	I	10	CEA	CD28, CD3z	Cy	[34]
Breast cancer	0	6	MET	4-1BB, CD3z	None	[35]
Neuroblastoma	I	11	GD2	CD28, OX40, CD3z	None, Flu/Cy, Flu/Cy+PD-1 inhibitor	[36]
Glioblastoma	I	10	EGFR	4-1BB, CD3z	None	[37]
Glioblastoma	I	17	HER2	CD28, CD3z	None	[38]
Biliary tract cancer and pancreatic carcinoma	I	11	HER2	4-1BB, CD3z	Cy/nab-paclitaxel	[39]
Pancreatic ductal adenocarcinoma	I	6	mesothelin	4-1BB, CD3z	None	[40]
Mesothelioma, Ovarian carcinoma, Pancreatic ductal carcinoma	I	15	mesothelin	4-1BB, CD3z	None or Cy	[28]
Prostate carcinoma, pancreatic carcinoma	I	13	PSMA	4-1BB, CD3z + TGFBDN	None or Flu/Cy	[29]
Mesothelin+ Solid tumor	I	27	mesothelin	CD28, CD3z	None, Cy, Cy+ PD-1 inhibitor	[41]
Glioblastoma	I	18	EGFR	CD28, 4-1BB, CD3z	Flu/Cy + IL-2	[42]
Liver metastases	Ib	6	CEA	CD28, CD3z	Selective intra-arterial radiation with SIR spheres	[43]
Hepatocellular carcinoma	I	13	GPC3	CD28, CD3z	Cy alone or Flu/Cy	[44]
Neuroblastoma	I	17	GD2	CD28, CD3z	None, Cy, Flu/Cy	[45]
Mesothelin+ Solid tumor	I	15	mesothelin	CD28, CD3z	None	[46]
Neuroblastoma	I	3	GD2	CD28, CD3z + IL15	Flu/C	[47,48]

* Cy: Cyclophosphamide; Flu: Fludarabine; SIR: Selective internal radiation.

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
