# Peer review of "Chimeric Antigen Receptor Immunotherapy for Solid Tumors: Choosing the Right Ingredients for the Perfect Recipe"

_cancers, 2022, doi:10.3390/cancers14215351_

Round 1

Reviewer 1 Report

The review is well written and the reader can get a general description of all the variables implicated in CAR T cell therapy design and conceptualization. However, I would suggest to make a more detailed focus on the solid tumor context, better describing preclinical and clinical studies when they are introduced in a particular session.

Here below some comments:

-        382-387 not exactly true, non viral transposon are known to display a high-cargo capacity, please correct or better discuss this point;

-        When mentioning the clinical studies, try to add results obtained and not only indicating a list of trials exploiting a specific strategy described (i.e.  390 already used in CAR clinical trials in the hematology and showed encouraging results Please explain what do you mean by encouraging results, safety, efficacy as compared to viral CAR T?;

-          same comment for line 425, a brief description of the referred study would be readily available for the reader instead of having the sole reference to the study.

-          Using tandem CAR or utilizing constructs coding for 508 multiple fully-functional CAR, several groups have shown a broader ability of engineered 509 cells to recognize and kill tumor cells [110,111]. This approach has already reached clinical 510 testing in the hematology fields with encouraging results and is currently under evalua- 511 tion in solid tumors [24,112,113]

Please, describe these studies. The review is on CAR T for solid tumors, so the reader expects to have information on the solid targets and the results obtained so far.

553-554: which report? Please specify

-          The TME paragraph I think is one of the most interesting ones in this context of CAR T cells for solid tumors. I would shrink some paragraph that are more general and can be found in any review on CAR T cell therapy (Gene engineering tools for example) and I would extend the ones more related to the solid tumor targeting by CARs.

Reviewer 2 Report

In this extensive review, Castiello et al. have reviewed the most promising approaches to chimeric antigen receptor T cell therapies under development in the solid tumour. They have discussed their specific advantages and disadvantages. The literature and references are fine and it seems to me that the authors have reviewed and prepared a really nice and informative manuscript. 

I have to thank and congratulate the authors for bringing together this information and I recommend this manuscript for publication in Cancer.
